# Modelling the Cost-Effectiveness and Budget Impact of a Newborn Screening Program for Spinal Muscular Atrophy and Severe Combined Immunodeficiency

**DOI:** 10.3390/ijns8030045

**Published:** 2022-07-20

**Authors:** Sophy T. F. Shih, Elena Keller, Veronica Wiley, Michelle A. Farrar, Melanie Wong, Georgina M. Chambers

**Affiliations:** 1Surveillance, Evaluation and Research Program, Kirby Institute, University of New South Wales, Sydney, NSW 2052, Australia; 2National Perinatal Epidemiology and Statistics Unit, Centre for Big Data Research in Health and School of Clinical Medicine, University of New South Wales, Sydney, NSW 2052, Australia; e.keller@unsw.edu.au (E.K.); g.chambers@unsw.edu.au (G.M.C.); 3NSW Newborn Screening Programme, Children’s Hospital Westmead, Westmead, NSW 2145, Australia; veronica.wiley@health.nsw.gov.au; 4Faculty of Medicine and Health, University of Sydney, Sydney, NSW 2006, Australia; 5Department of Neurology, Sydney Children’s Hospital, Randwick, NSW 2031, Australia; m.farrar@unsw.edu.au; 6Discipline of Paediatrics, School of Clinical Medicine, UNSW Medicine and Health, UNSW Sydney, Sydney, NSW 2052, Australia; 7Department of Allergy and Immunology, Children’s Hospital at Westmead, Westmead, NSW 2145, Australia; melanie.wong@health.nsw.gov.au

**Keywords:** SMA, SCID, newborn screening, cost-effectiveness, Markov model, budget impact analysis

## Abstract

Spinal muscular atrophy (SMA) and severe combined immunodeficiency (SCID) are rare, inherited genetic disorders with severe mortality and morbidity. The benefits of early diagnosis and initiation of treatment are now increasingly recognized, with the most benefits in patients treated prior to symptom onset. The aim of the economic evaluation was to investigate the costs and outcomes associated with the introduction of universal newborn screening (NBS) for SCID and SMA, by generating measures of cost-effectiveness and budget impact. A stepwise approach to the cost-effectiveness analyses by decision analytical models nested with Markov simulations for SMA and SCID were conducted from the government perspective. Over a 60-year time horizon, screening every newborn in the population and treating diagnosed SCID by early hematopoietic stem cell transplantation and SMA by gene therapy, would result in 95 QALYs gained per 100,000 newborns, and result in cost savings of USD 8.6 million. Sensitivity analysis indicates 97% of simulated results are considered cost-effective against commonly used willingness-to-pay thresholds. The introduction of combined NBS for SCID and SMA is good value for money from the long-term clinical and economic perspectives, representing a cost saving to governments in the long-term, as well as improving and saving lives.

## 1. Introduction

Spinal muscular atrophy (SMA) is a devastating genetic neuromuscular disorder, with an incidence of approximately one in 11,000 live births across the major ethnic groups. It results in significant disability and mortality and has been the leading genetic cause of infant death before the introduction of disease-modifying therapies that increase the survival motor neuron (SMN) protein. The severity of SMA ranges from progressive infantile paralysis and premature death (type I) to onset in adulthood with limited motor neuron loss and normal life expectancy (type IV) [1]. Nusinersen, an antisense oligonucleotide that modulates survival motor neuron 2 (*SMN2*) pre-mRNA splicing to increase SMN protein levels, is administered by intrathecal injection every four months after an initial loading period. Onasemnogene abeparvovec, or SMA gene therapy, delivers a functional human *SMN* cDNA transgene to cells via a recombinant adeno-associated virus with the ability to cross the blood–brain barrier following a single intravenous injection. The most beneficial response to treatment to date has been seen in infants treated prior to symptom onset [2], with the prospect of normal motor development during infancy and childhood, highlighting the value of early diagnosis.

Primary immune deficiency (PID) is a group of disorders in which there is a defective defense against infection, with the most severe forms, such as severe combined immunodeficiency (SCID), characterized by the absence of both humoral and/or cellular immunity. SCID is a rare, inherited disease affecting between 1 in 50,000 to 1 in 100,000 newborns. Two-thirds of the cases are X-linked, meaning that it predominantly occurs in males [3]. While affected babies typically show no signs of the disease at birth, without treatment they often do not survive past the first two years of life, due to recurrent infections. Viral and fungal infections tend to occur from the age of two months, and bacterial infections from the age of four to six months, when the trans-placentally acquired maternal antibodies have disappeared. Respiratory tract and gastrointestinal infections are most common, leading to malnutrition and impaired growth [4]. Approximately 50% of the children with SCID present with a life-threatening respiratory tract infection secondary to *Pneumocystis jirovecii* (PJP) infection, typically between 2 to 6 months of age [5]. The infants with SCID suffer recurrent, often devastating, infections, with a high risk of morbidity and mortality if not recognized and treated early. A mortality rate of 35% has been reported in immunocompromised children with PJP infection [6]. The most common treatment is hematopoietic stem cell transplantation (HSCT), from a related or unrelated registry-derived donor. The survival rate after HSCT is highest in the infection-free infants treated within the first 3.5 months of life, with 94% of patients alive after two years. In older infants, survival is mainly determined by the absence of infections, with 90% of infection-free infants alive two years after HSCT, compared to 82% in those with resolved infections and 50% in those with active infections during transplantation [7].

Newborn screening (NBS) has the potential to enable the early diagnosis of both SMA and SCID, using a multiplex PCR assay on DNA extracted from a single punch of the routine dried blood spot (DBS) sample. Importantly, the benefits of early diagnosis and initiation of treatment are now increasingly recognized, with the most beneficial response to treatment to date, in these disorders, seen in the patients treated prior to the onset of symptoms. SCID has been included in many NBS programs globally, including those within all of the US states and some European countries. Since August 2018, the New South Wales (NSW) and Australian Capital Territory (ACT) NBS program moved with the international trend of screening for PID, and for the first time included testing for SMA and PID as a pilot program.

Previous studies of NBS for SCID have suggested that NBS for SCID is cost-effective [3,8,9,10,11,12,13]. Many health technology assessment and modelled cost-effectiveness analyses have been conducted to assess the clinical and economic values of novel disease-modifying treatment for SMA [14,15,16,17,18,19,20]. A small number of studies evaluated the cost-effectiveness of universal NBS for SMA [21,22]. Even so, the economic appraisal of a combined NBS program for SCID and SMA is lacking. We previously reported the cost-effectiveness of introducing NBS for SMA with early treatment of gene therapy or nusinersen [22] and NBS for SCID with early HSCT separately (IJNS, companion manuscript). However, the cost-effectiveness of NBS for SCID and SMA together is unknown and the budget impact for governments to add these two conditions to routine NBS has not been assessed. There is a clear rationale for combining NBS for SMA alongside SCID purely on a cost-of-screening basis, because adding SMA to the SCID panel only adds an extra AUD 1 to the existing AUD 7 test. However, the costs of treating these conditions, SMA in particular, are very high and may be needed over a lifetime. Therefore, the economic argument of whether to introduce NBS for these conditions necessitates an evaluation that incorporates subsequent treatment with new disease-modifying therapies to determine if their addition is cost-effective and represents ‘value for money’, from the Australian government perspective.

Therefore, to inform future policy development in the Australian context, our objectives were to evaluate the cost-effectiveness and financial impact of introducing a multiplex assay for the NBS of SMA and PID to the current NBS program, from the Australian government’s perspective as the payer.

## 2. Materials and Methods

The NSW/ACT NBS program initiated a state-wide pilot for SMA and PID in August 2018. Heel-prick blood samples on filter paper (Guthrie Card) were collected within 48–72 h of birth and sent to the NSW/ACT Newborn Screening Laboratory for analysis. The real-time PCR 4-plex assay directly amplified the target and control DNA sequences extracted from a single 3.2 mm DBS punch per sample, using reagents specific for *SMN1*, T-cell receptor excision circles (TREC), kappa-deleting recombinant excision circles (KREC), and a reference gene (RNAseP). The number of *SMN2* copies was measured as a second-tier screen by droplet digital PCR in those with 0 SMN1. During the pilot study, all of the infants with 0 *SMN1* were screen positive and infants with four or more copies of *SMN2* were included.

Instead of modelling all of the PIDs, the current analysis focused on SCID, which is the most severe form of PID. Informed by our state-wide SMA and PID NBS pilot program and available literature, decision analytical models nested with Markov simulation models for each of the treatment strategies (two for SCID and four for SMA) were constructed. We developed a stepwise approach by undertaking three sets of cost-effectiveness analyses, presented in Figure 1. Firstly, we conducted the cost-effectiveness analysis of treating SCID pre-symptomatically with HSCT versus treating surviving SCID with HSCT after clinical diagnosis using Markov simulations (Figure 1; orange compartment). Similarly, the cost-effectiveness analyses of the pairwise comparisons between alternative treatments for SMA were performed (Figure 1; blue compartment). Secondly, we combined the analyses using a decision analytic model (with embedded Markov treatment strategies, Figure 2) to evaluate the cost-effectiveness of the combined NBS strategy for SCID and SMA (Figure 1; green compartment).

All of the model simulations were performed using the TreeAge Pro 2020 software (TreeAge Pro 2022 software, Williamstown, MA, USA). Details of the Markov cohort simulation and model parameter values and estimates were described previously [22] (IJNS companion manuscript) and the model parameters are presented in Appendix A. In summary, we defined 11 health states for SMA with the length of each Markov cycle of 6 months, based on the clinical observations of motor milestone development in the SMA type 1, 2, and 3 cohorts [22]; and six health states for SCID with a Markov cycle length of three months to capture the distinctions in disease progression and the natural history between early and late diagnosis in the first year of life, with consideration of family history and mortality before receiving HSCT (IJNS, companion manuscript). The modelled health outcomes by quality-adjusted life-years (QALYs), life-years (LYs), and costs were used to evaluate the NBS strategies and candidate therapies in terms of the short-term and long-term cost-effectiveness by calculating an incremental cost-effectiveness ratio (ICER). A probabilistic sensitivity analysis (PSA), using a Monte Carlo simulation, was performed to generate cost-effectiveness planes and to derive 95% confidence intervals (CI). One-way sensitivity analysis of the key model parameters was presented in a Tornado diagram.

All of the analyses were conducted from the government’s perspective, using 5-year and 60-year time horizons. The choice of the 5-year time horizon was supported by available clinical data on the treatment effects of the new disease-modifying therapies to provide certainty for the results [23]. The 60-year time horizon was applied to account for the long-term costs and outcomes. The QALYs and costs were discounted at 3% per annum. The budget impact analysis was undertaken to inform the affordability of the new NBS program, from a government perspective, with a combined SMA and SCID test for the next 5 years. The net budget impact was estimated for the introduction of combined SCID and SMA to the current NBS.

The direct healthcare costs were considered in the analysis, including screening, diagnosis, treatment, and other medical care. The costs were sourced from our pilot program, reimbursements from Australian Pharmaceutical Benefits Scheme and Medicare Benefits Schedule, literature, and the assumptions of gene therapy costs based on overseas market prices (not publicly available in the Australian market). The details of the cost assessments can be found from the published studies (IJNS, companion manuscript) [22]. The costs were estimated in USD and reported for the reference year 2018 with adjustments of the Consumer Price Index (CPI) and Purchasing Power Parity (PPP) conversions, published by the OECD [24].

This study was approved by The Sydney Children’s Hospitals Network Human Research Ethics Committee (reference number LNR/18/SCHN/307). The parents of the children participating in the NBS pilot evaluation study provided written consents.

## 3. Results

From August 2018 to August 2020, in total 202,388 newborns were screened for SMA and PID, specifically SCID and B-cell deficiency [25,26]. There were 18 potential SMA and 114 potential PID cases detected during the pilot screening period, and further samples were requested for diagnosis where 17 SMA and 8 PID cases were proven [26,27].

### 3.1. Cost-Effectiveness Analysis of NBS for SMA

The details of the cost-effectiveness analyses of the four treatment strategies and NBS for SMA alone, from the societal perspective that included informal care and parents’ loss of productivity in addition to direct healthcare costs, were reported in Shih et al. [22]. Here, we present the summary results of NBS for SMA with gene therapy, compared to nusinersen treatment without NBS from the government’s perspective that only took account of the direct healthcare costs to align with information on the government’s policy and the impact on their budgets. In the short-term, over the 5-year time horizon, treating one infant diagnosed with SMA before symptom onset with gene therapy in comparison to the late initiation of nusinersen in clinically diagnosed SMA (current practice in Australia), would incur an additional USD 487,000 in healthcare costs and achieve 0.78 QALY gains (ICER USD 621,000/QALY). In the long-term, in comparison to current practice, gene therapy in screen-detected SMA would be dominant, achieving 9.93 QALYs gained and saving USD 1.06 million over a 60-year projection.

Over a 5-year time horizon, compared to no screening and late nusinersen treatment, the NBS for SMA with pre-symptomatic gene therapy would gain seven QALYs at USD 4.7 million per 100,000 infants, resulting in an ICER of USD 697,000/QALY from the Australian governmental perspective. Over the 60-year time horizon, compared to late nusinersen treatment without screening, screening every newborn in the population and treating the screen-detected SMA patients with gene therapy would be dominant (saving USD 8.4 million with 85 QALYs gained), with a 95% CI of the ICER ranging from dominant to USD 66,500/QALY.

The probabilistic sensitivity analysis illustrated that 96% of the simulated ICER results for NBS with gene therapy in the long-term fell below the threshold of USD35,000/QALY (AUD 50,000/QALY) or were dominant, demonstrating a high probability of QALY gains (positive incremental effectiveness) with a cost saving (negative incremental cost).

### 3.2. Cost-Effectiveness Analysis of NBS for SCID

Details of the cost-effectiveness of NBS for SCID alone were reported in Shih et al. (IJNS companion manuscript) and a summary of the modelling ICER results are presented here. Assessing the treatment strategies in the short term, the pre-symptomatic treatment of a SCID patient with early HSCT dominated over late treatment with HSCT. Early HSCT resulted in 1.53 more QALYs per child diagnosed with SCID, at a saving of USD 123,000 over a 5-year time horizon. Over a 60-year time projection, early treatment with HSCT also dominated over late treatment, with 6.14 more QALYs gained per patient diagnosed with SCID, at savings of USD 137,000 from the government’s perspective.

In terms of the cost-effectiveness of NBS to detect infants with SCID and treated by early HSCT, the costs and QALYs for one newborn screened in the population resulted in an ICER of USD 144,000/QALY over 5 years. Over the 60-year time horizon, screening every newborn in the population and treating the diagnosed SCID with early HSCT, would result in more QALYs gained at a marginally higher cost, with an ICER of USD 33,600/QALY from the government’s perspective. With reference to the common willingness-to-pay thresholds of USD 35,000/QALY (AUD 50,000/QALY) in the Australian setting, this would be considered good value for money in healthcare.

Varying the costs and probabilities using a PSA produced a 95% CI of USD 17,900/QALY to USD 59,400/QALY, with approximately half of all of the simulations falling under the threshold of USD 35,000/QALY.

### 3.3. Cost-Effectiveness of the Addition of SCID and SMA Together into NBS Programs

The discounted costs, QALYs, Lys, and ICERs from the government’s perspective of a combined introduction of SCID and SMA into the current NBS panel, with the early treatment of screen-detected SMA by gene therapy and SCID by early HSCT, are presented in Table 1.

Over a 5-year time horizon, screening every newborn in the population and treating diagnosed SCID by early HSCT and SMA by gene therapy would cost USD 125 and result in 0.00013 QALYs and 0.00033 LYs per newborn screened in the population. Compared to the symptomatic SCID treated with late HSCT and clinically detected SMA with the late initiation of nusinersen, the NBS for SCID and SMA produced an additional 0.00009 QALYs and 0.00009 LYs at a cost of USD 46 for every newborn screened (nine QALYs/nine LYs gained at cost of USD 4.6 million in 100,000 newborns). Therefore, the ICER of NBS for SCID (early HSCT) and SMA (gene therapy) would be USD 496,000/QALY and USD 515,000/LY, albeit with wide 95% CIs from being dominant to USD 953,000/QALY and USD 1,145,000/LY, indicating substantial uncertainty around the estimates in the short term.

Over the 60-year time horizon, screening every newborn in the population for SMA and SCID together and treating diagnosed SCID by early HSCT and SMA by gene therapy, would result in incrementally more QALYs/LYs gained, at a lower cost compared to no NBS for these conditions. For each newborn screened in the population, 0.00095 QALYs/0.00137 LYs are gained coupled with cost savings of USD 86 from the Australian government’s perspective. This means, over the 60-year time projection, screening and early treatment with HSCT for SCID and gene therapy for SMA dominated over late treatment with HSCT and nusinersen, resulting in 95 QALYs/137 LYs gained and savings of USD 8.6 million per 100,000 newborns, from the government’s perspective.

The marginal analysis of adding NBS for SMA to an existing NBS program with SCID (SMA + SCID vs. SCID) produced similar results to those of adding combined SMA and SCID to the NBS panel.

### 3.4. Sensitivity Analyses

The sensitivity analyses indicated that the incidence of SMA and SCID, as well as the cost of the new disease-modifying therapies (nusinersen and gene therapy) were the significant parameters influencing the cost-effectiveness results, as shown in the Tornado diagram in Figure 3.

The results of the PSA for the NBS cost-effectiveness of combined SCID and SMA over 60 years, using 1000 iterations of Monte Carlo simulations of costs and effectiveness, are presented in Figure 4. This graph presents total QALYs and total costs for each of the 1000 simulations of the current NBS program without SCID and SMA and the future NBS program with SCID and SMA. The incremental cost-effectiveness plane in Figure 5 demonstrates a strong case for the future NBS panel to include SCID and SMA. In Figure 5, the PSA shows high confidence of this conclusion, with 97% of the simulated iterations falling below a threshold of USD 35,000/QALY (AUD 50,000/QALY), or in the lower-right quadrant (more health gains and cost savings). This suggests that the combined NBS for SCID and SMA, together with early treatment with HSCT and gene therapy, respectively, is highly cost-effective and robust to various assumptions.

### 3.5. Budget Impact on Government Accounts of a Combined Addition of SCID + SMA into a NBS Program

By screening SCID and SMA and treating screen-detected cases with early HSCT and gene therapy in a cohort of 100,000 newborns, total expected costs of USD 17 million over 5 years are estimated (undiscounted nominal value) (Table 2). The total expected costs of current NBS without SCID and SMA and treating clinically diagnosed cases with late HSCT and nusinersen over 5 years are estimated to be USD 12 million (undiscounted nominal value). By introducing the SCID and SMA screening to the current NBS program, an additional USD 4.5 million over 5 years would be required from the governmental budgets to screen a cohort of 100,000 newborns and to treat the diagnosed SCID and SMA cases with early HSCT and gene therapy. Screening costs of USD 558,000 occurred in the first year and only accounted for 3% of the overall budgets over 5 years.

The total budget over a period of 5 years for future NBS including SCID and SMA would cost USD 80 million to screen and treat the screen-detected cases with gene therapy and early HSCT; compared to a total budget over 5 years without NBS for SCID and SMA of USD 45 million to treat the clinically detected cases with late HSCT and nusinersen (current practice in Australia; Table 3). An additional USD 35 million would be required over the first 5 years of operation for the future NBS program, mainly to fund early one-off treatment with gene therapy, which is estimated to cost USD 1.54 million per child.

## 4. Discussion

Multiplex real-time PCR assay has been used in universal newborn screening for SCID and SMA [28,29]. However, the real-world data, particularly regarding economic evidence of NBS programs using the multiplex PCR assay, are lacking. This is the first study to evaluate the combined addition of SCID and SMA assays to population NBS programs. Our analysis is the first of its kind to provide cost-effectiveness and budgetary implications for policy development in Australia, but the evidence is also relevant for other jurisdictions. Our modelled cost-effectiveness results suggest that introducing the combined assay of SMA and SCID provides good value for money, not only saving lives but also saving costs in the long-term. Compared to the current NBS in Australia, NBS for SMA and SCID with early treatment by gene therapy and HSCT, respectively, is a dominant strategy in economic terms. The cost-effectiveness results over 60 years, when the long-term benefits of health and costs are realized, demonstrate strong economic benefits of the future NBS panel to include a combined SCID and SMA screening and treatment with early HSCT and gene therapy in pre-symptomatic babies.

However, the ICERs of the NBS for SCID and SMA combined in the short-term does not show to be cost-effective, even with a special consideration of the rare/ultra-rare condition willingness-to-pay thresholds ranging from USD 100,000 to USD 150,000 per QALY recommended by the Institute of Clinical and Economic Review [30]. The ICER of nearly half a million USD per QALY in the short-term is driven by the highly expensive one-dose administration of gene therapy for SMA. Our marginal analyses also indicate that the overall results are driven by NBS for SMA, because SMA has four–five times the incidence of SCID, and gene therapy is highly dominant in terms of costs.

With the first ever disease-modifying treatments reaching SMA patients, and thus revolutionizing clinical practice, the landscape for patients with SMA, and their families, has changed irrevocably. Based on the AUD 2 million (USD 1.54 million) price of gene therapy used in our base-case evaluation, gene therapy is likely to be considered a highly cost-effective treatment strategy for the newborn screen-detected SMA. Even with the full market cost of gene therapy at AUD 2.73 million (USD 2.1 million), the ICER for NBS with gene therapy would change from the dominant to USD 21,000/QALY, which is still considered cost-effective in the Australian healthcare setting, in the comparison of gene therapy in screened SMA over the current practice of nusinersen therapy without NBS [22]. Our cost-effectiveness results for NBS with gene therapy compared to no NBS were in line with other published modelling studies in the United States by conference abstracts, indicating that NBS with gene therapy is likely to be cost-effective [31,32]. In the studies examining universal newborn screening for SMA, it has been found that NBS and treatment with nusinersen was not cost-effective, even against higher thresholds for rare diseases, mainly due to the required ongoing treatment maintenance by nusinersen injections [21,32].

For almost 20 years, until the addition of screening for congenital adrenal hyperplasia in some Australian states from 2018, additions to the established Australian NBS program were not forthcoming, despite significant advances in the availability of treatments for a number of serious genetic illnesses. New technologies and approaches to diagnosis and care allow clinicians, patients, and caregivers to access novel, life-changing therapies for diseases that, until now, have received only supportive care. SMA represents one such disease and its approved genetic therapies have provided a new paradigm for the treatment of SMA—moving from supportive care to interventional care.

Our study results provide timely and critical information for the potential future national adoption of DNA-based NBS, to further expand screening programs from the current NSW/ACT pilot of NBS for SMA and PID. While our evaluation only focusses on SCID, the pilot NBS program was capable of screening for a number of PIDs. The cost–benefit would be improved if other treatable immune deficiencies were included in the modelled health outcomes. In terms of adopting this into an Australian nationwide screening program, the recent pilot study within the NSW/ACT NBS Program demonstrated its capacity to undertake the NBS of SMA and SCID in a highly reproducible manner, in line with the established standards, and can be effectively translated into practice [33]. Our evaluation also provides useful information for other countries that plan to establish or adapt their screening procedures for SMA and SCID.

The NSW/ACT pilot for SMA and PID proved that early treatment of SMA enables the achievement of early, age-appropriate motor milestones. The pilot study has assessed the practice and policy requirements, resulting in the Australian Department of Health’s recommendation for inclusion of SMA in NBS nationally. The results from the pilot study have demonstrated the accuracy, efficiency, and short-term health outcomes of the screening and diagnostic pathway [25]. New genetic therapies for SMA have attained regulatory approvals and reimbursements. The perspectives of parents and clinicians involved in the NBS pilot were evaluated in a prospective mixed-methods study and affirmed the acceptability, sustainability, and utility of the NBS for SMA [26]. Our health economic model of the NBS showed that the lifetime cost of treating SMA can be reduced with NBS, improving the quality and length of life of infants with SMA at a universally acceptable threshold for value-for-money in healthcare with gene therapy [22].

The patterns of budget distribution over 5 years are distinct between future NBS and current NBS, with larger upfront commitments required for NBS for SCID and SMA due to early treatment with gene therapy for SMA costing USD 1.54 million per screen-detected case. At this level of gene therapy price, the net budget requires an additional USD 35 million over 5 years to screen and treat early SMA and SCID. However, over a longer time horizon, NBS for SCID and SMA would be cost-saving from a government perspective, as the gap in the cost between the future and current NBS is narrowing when the total cost without NBS for SMA and SCID increases over time, due to ongoing nusinersen treatment for SMA. In a scenario analysis, with the gene therapy price dropped to USD 1 million per dose (AUD 1.45 million), the net cost of the future NBS with SMA and SCID would become lower than the current NBS without SMA and SCID from the 5th year. It is also worth noting that the NBS programs are operated by the state governments in Australia, while medical care services are provided by the federal government through Australia’s universal health insurance scheme, called Medicare. The total screening costs are just over half a million USD, which only accounts for a small fraction (3%) of the total budgets required of USD 17 million for 100,000 babies in the first 5 years. With greater purchasing power from a federal government (e.g., the Australian Commonwealth government) to negotiate the novel high-cost disease-modifying therapies (e.g., gene therapy), the likelihood of a better investment return would be increased. Thus, early investment in screening and gene therapy and HSCT indicates greater financial returns to the government in the long-term, with NBS and the early initiation of disease-modifying therapies becoming cost-saving, compared to the current NBS program, after 5 years. The future Australian clinical practice of SMA treatment may include the reimbursement of gene therapy, either in pre-symptomatic cases with NBS or in cases with clinical manifestation without NBS. The economic value of NBS for SMA should be re-assessed when real-world clinical evidence becomes available. Nevertheless, the potential cost-effectiveness for such a scenario is anticipated to be not much worse than our current comparison, as the incremental cost would be the screening costs (half a million USD in the current evaluation), while the treatment costs (i.e., gene therapy) would be cancelled out between NBS and no NBS, and NBS would result in more QALYs than no NBS. In Australia, the reimbursement of disease-modifying therapies is determined by the Pharmaceutical Benefits Scheme and does not permit combination/add on therapy. The assumption we made in the model of a one-off dosage of gene therapy may be underestimated, compared to the clinical practice in other jurisdictions.

We evaluated and validated both the SMA and SCID Markov models with published studies to calibrate the models. Although with endeavor to model the costs and outcomes to represent the clinical reality, our study is not free from limitations, in particular the need to model over a life-long timeframe while only short-term clinical data are available. It was inevitable to assume constant transition probabilities between health states over the entire Markov process, implying treatment benefits continued beyond the available observed outcomes. The caveat of such an assumption needs careful consideration in utilizing the modelled results. The quality of life measurement is challenging in an evolving treatment landscape, let alone the difficulties in measuring quality of life in very young children. The prospective data of quality of life in individuals treated by novel therapeutic interventions for SMA are lacking, and thus cross-sectional surrogate utility values from older SMA patients matched by phenotypes were used. This may underestimate the QALYs gain, as significant motor function development has been observed in young patients treated by the disease-modifying therapy. Our results are conservative in favor of non-screening because we conducted our analysis from a governmental healthcare payer’s perspective and, thus, we did not include the societal costs associated with loss of productivity. Furthermore, the SMA medical costs in the intervention group of our model are likely to be over-estimated, because we applied higher medical costs estimated before the introduction of the disease-modifying therapy [34]. Another limitation of our analysis is the lack of precision in forecasting the budgetary impact. While the budget impact analysis provides future resourcing estimates from a government perspective, the estimates become less precise over longer projection periods. Such uncertainties are not easily parameterized, cannot be meaningfully quantified, and thus should be viewed with caution [35].

## 5. Conclusions

The findings of our analyses suggest that a combined introduction of NBS for SCID and SMA provides good value for money from the long-term clinical and economical perspective, saving lives and costs and improving the quality of life of children and families affected by these conditions.

## Figures and Tables

**Figure 1 IJNS-08-00045-f001:**
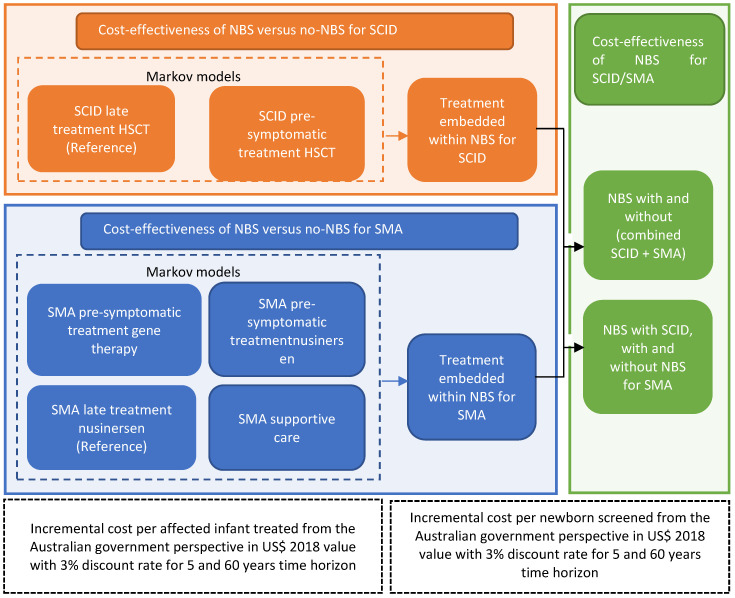
Analytical framework for SCID and SMA newborn screening and treatment strategies.

**Figure 2 IJNS-08-00045-f002:**
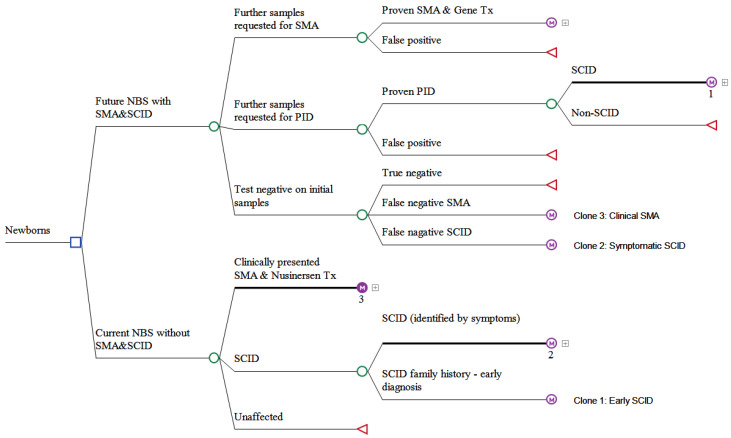
Schematic presentation of the decision analytical model comparing future NBS program adding combined SMA and PID with current NBS program without SMA and PID.

**Figure 3 IJNS-08-00045-f003:**
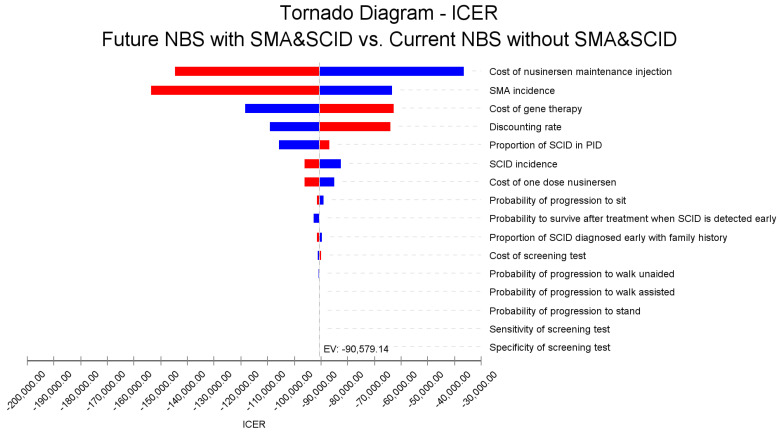
Tornado diagram of the one-way sensitivity analysis showing the impact of key parameters on the ICER for combined NBS for SCID and SMA, government perspective, 60 years, USD 2018. Note: Red bars indicate an increase in the parameter value from the base case value 257 (Expected Value, EV line) and blue bars show otherwise.

**Figure 4 IJNS-08-00045-f004:**
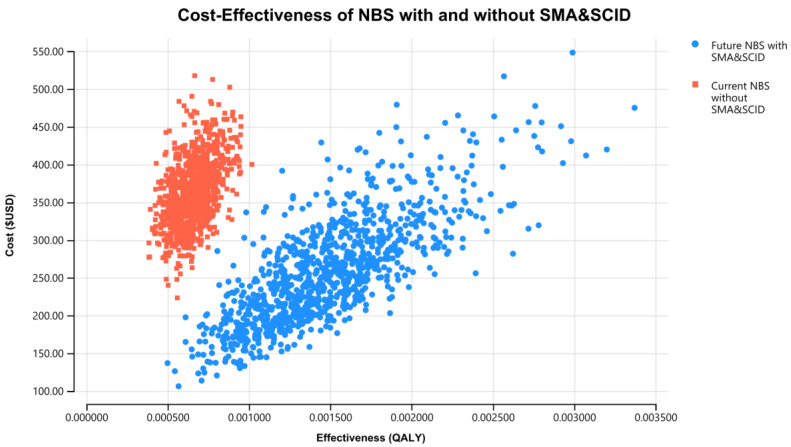
Total cost and total QALYs for combined NBS for SCID and SMA treated with early HSCT and gene therapy and no screening for SCID and SMA, government perspective, 60 years, USD 2018.

**Figure 5 IJNS-08-00045-f005:**
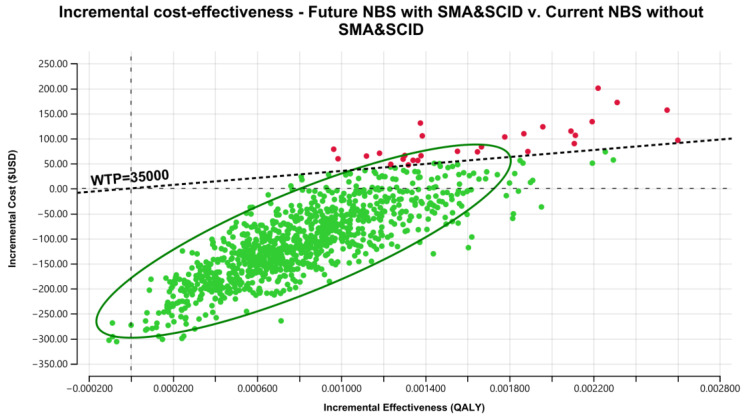
Incremental cost and incremental QALYs of a combined NBS panel for SCID and SMA compared to no screening, government perspective, 60 years, USD 2018. (Note: Green dots represent iterations considered cost-effective (ICER less than the willingness-to-pay (WTP) threshold $USD 35,000/QALY); while red dots represent iterations considered not cost-effective (ICER greater than the WTP threshold).

**Table 1 IJNS-08-00045-t001:** Cost-effectiveness of a combined NBS for SCID and SMA treated with early HSCT and gene therapy over 5 and 60 years from the government’s perspective, discounted at 3% p.a. (US$ 2018).

Strategy	Cost (US$)	Incremental Cost (US$)	QALY	Incremental QALY	ICER (US$/QALY)
5 Years	(95% CI)	(95% CI)	(95% CI)	(95% CI)	(95% CI)
Current NBS without SMA and SCID	$124.72 ($97.74, $156.36)		0.00013 (0.00010, 0.00018)		
Future NBS Add SMA and SCID	$170.33 ($104.90, $254.52)	$45.61 (−$25.96, $136.88)	0.00022 (0.00015, 0.00032)	0.00009 (0.00002, 0.00018)	$495,506 (dominant, $952,608)
60 Years	(95% CI)	(95% CI)	(95% CI)	(95% CI)	(95% CI)
Current NBS Without SMA and SCID	$362.58 ($285.53, $459.35)		0.00077 (0.00056, 0.00099)		
Future NBS Add SMA and SCID	$276.59 ($160.64, $441.65)	−$86.00 (−$230.88, $82.59)	0.00172 (0.00103, 0.00263)	0.00095 (0.00029, 0.00182)	Dominant (dominant, $46,753)
**Strategy**	**Cost** (US$)	**Incremental Cost** (US$)	**LY**	**Incremental LY**	**ICER** (US$/LY)
5 Years	(95% CI)	(95% CI)	(95% CI)	(95% CI)	(95% CI)
Current NBS without SMA and SCID	$124.72 ($97.74, $156.36)		0.00033 (0.00027, 0.00040)		
Future NBS Add SMA and SCID	$170.33 ($104.90, $ 254.52)	$45.61 (−$25.96, $136.88)	0.00042 (0.00029, 0.00061)	0.00009 (−0.00006, 0.00028)	$514,844 (dominant, $1,145,110)
60 Years	(95% CI)	(95% CI)	(95% CI)	(95% CI)	(95% CI)
Current NBS without SMA and SCID	$362.58 ($285.53, $459.35)		0.00130 (0.00108, 0.00155)		
Future NBS Add SMA and SCID	$276.59 ($160.64, $441.65)	−$86.00 (−$230.88, $82.59)	0.00267 (0.00164, 0.00397)	0.00137 (0.00035, 0.00268)	Dominant (dominant, $37,280)

QALY: Quality-adjusted life-year; LY: Life-year; ICER: Incremental cost-effectiveness ratio.

**Table 2 IJNS-08-00045-t002:** Costs of future NBS with screening for SCID and SMA and treating with early HSCT and gene therapy and current NBS without screening with late HSCT and nusinersen treatment for a cohort of 100,000 newborns (undiscounted US$ 2018 value).

Future NBS	No. of Newborns (N)	Year 1 (US$)	Year 2 (US$)	Year 3 (US$)	Year 4 (US$)	Year 5 (US$)	Total (US$)
Screening *	99,989	$558,371					$558,371
Screened SMA	8.5	$13,759,906	$518,371	$390,943	$357,220	$341,565	$15,368,005
Clinical SMA	0.6	$318,122	$140,049	$125,867	$110,384	$97,913	$792,335
Screened SCID	2	$245,120	$7750	$6870	$6097	$5413	$271,250
late SCID	0.01	$2388	$58	$51	$45	$40	$2581
Total budget	100,000	$14	$666,227	$523,731	$473,746	$444,931	$16,992,542
**Current NBS**	**No. of Newborns** (N)	**Year 1** (US$)	**Year 2** (US$)	**Year 3** (US$)	**Year 4** (US$)	**Year 5** (US$)	**Total** (US$)
No Screening *	99,989	$0					
Clinical SMA	9.1	$4,824,852	$2,124,069	$1,908,986	$1,674,157	$1,485,020	$12,017,084
Clinical SCID	1.6	$382,024	$9264	$8167	$7203	$6357	$413,014
Early SCID by family history	0.4	$49,024	$1550	$1374	$1219	$1083	$54,250
Total budget	100,000	$5,255,900	$2,134,883	$1,918,527	$1,682,579	$1,492,460	$12,484,349

* Unaffected newborns without SMA and SCID.

**Table 3 IJNS-08-00045-t003:** Expected 5-year total budgets of future NBS with combined screening for SCID and SMA and treatment with early HSCT and gene therapy, compared to current NBS without screening for SCID and SMA and treatment with late HSCT and nusinersen (undiscounted US$ 2018 value).

Future NBS	Year 1 (US$)	Year 2 (US$)	Year 3 (US$)	Year 4 (US$)	Year 5 (US$)	Total (US$)
Cohort 1	$14,883,907	$666,227	$523,731	$473,746	$444,931	$16,992,542
Cohort 2		$14,883,907	$666,227	$523,731	$473,746	$16,547,612
Cohort 3			$14,883,907	$666,227	$523,731	$16,073,865
Cohort 4				$14,883,907	$666,227	$15,550,134
Cohort 5					$14,883,907	$14,883,907
Total budget	$14,883,907	$15,550,134	$16,073,865	$16,547,612	$16,992,542	$80,048,061
**Current NBS**	**Year 1** (US$)	**Year 2** (US$)	**Year 3** (US$)	**Year 4** (US$)	**Year 5** (US$)	**Total** (US$)
Cohort 1	$5,255,900	$2,134,883	$1,918,527	$1,682,579	$1,492,460	$12,484,349
Cohort 2		$5,255,900	$2,134,883	$1,918,527	$1,682,579	$10,991,889
Cohort 3			$5,255,900	$2,134,883	$1,918,527	$9,309,310
Cohort 4				$5,255,900	$2,134,883	$7,390,783
Cohort 5					$5,255,900	$5,255,900
Total budget	$5,255,900	$7,390,783	$9,309,310	$10,991,889	$12,484,349	$45,432,231

## Data Availability

Results of the marginal analysis of adding NBS for SMA to an existing NBS program with SCID (SMA + SCID vs. SCID) are not shown but available from authors.

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
