# Peer review of "Modelling the Cost-Effectiveness and Budget Impact of a Newborn Screening Program for Spinal Muscular Atrophy and Severe Combined Immunodeficiency"

_2409-515X, 2022, doi:10.3390/ijns8030045_

Round 1

Reviewer 1 Report

L83-88: This requires clarification. The TREC assays used to screen for SCID worldwide are not specific to SCID but rather test for deficient T-cell function common to a number of PIDs. Therefore, it is misleading to suggest that the NSW/ACT approach differs from that of other NBS programs screening for SCID. Also, the following statement is imprecise. “Whilst SCID screening was in place since 2008, screening for SMA was not at all common in 2018.” The first state-wide pilot NBS projects for SCID were introduced in two US states in 2008 and the first state-wide pilot NBS projects for SMA were introduced in several US states in 2018. Neither SCID NBS was common in 2008 nor SMA in 2018. Screening for SCID was first recommended nationally in the USA in 2010 whereas SMA was first recommended in 2018.  

L88-89: This sentence appears to be contradicted by the sentence at L121-122.

L94-102: The authors indicate that they previously separately assessed the cost-effectiveness of NBS for SMA and SCID and that the present submission is the first analysis to jointly assess screening for SMA and SCID. It is not clear what the previous analyses did. Reference 22 indicates that the actual cost of SMA screening as implemented in NSW/ACT was US$5 per infant, which is equivalent to AU$7. The authors now say that the cost of adding SMA to the SCID assay is AU$1 and that the cost to screen for SCID is AU$7 per infant. If that is correct, reference 22 attributed the cost of screening for both SCID and SMA to SMA alone and made no mention of the fact that NSW/ACT started screening for both SMA and SCID in 2018 using a single assay. The previously published analysis presumably should have noted that “adding SMA to the SCID panel only adds an extra AU$1 to the existing AU$7 test.”  

L97-101: The authors do not provide a logical argument for the purported advantage in jointly modeling the cost of NBS for SCID and SMA. It is standard practice in health economic evaluation to assess incremental costs of an intervention. If there is an existing SCID assay, the cost of screening for SMA is the incremental cost of adding SMA to SCID. That has nothing to do with jointly modeling the cost of SMA and SCID screening. The previous US CEA of NBS for SMA modeled the cost of adding SMA to the SCID assay. There is no practical difference between a CEA of adding SMA using a multiplex assay for SMA and SCID and doing a joint CEA of screening for SCID and SMA.  

L126-128: Suggest that the authors clarify that they are referring to surviving infants receiving HSCT in the absence of NBS, since some infants with SCID die before they could receive the procedure.  

L174-177: References 25 and 26 document the SMA screening process and results but provide no information on PID screening as is implied. Please cite a source that documents PID screening or at a minimum acknowledge that references 25 and 26 only document SMA screening.

L179-186: The previous analysis assumed a much higher cost of SMA NBS, roughly 7 times higher than the present analysis. That needs to be explicitly acknowledged. A presumably less important difference is that the previous analysis reported costs from the societal perspective and the current analysis reports costs from the government payer perspective. The authors should explain exactly which costs are included in the societal perspective and how those differ from the government perspective. They should document the cost calculations in interest of transparency.

L186-188: The authors note a gain of 9.93 QALYs per infant with SMA comparing gene therapy in screened infants to nusinersen in clinically diagnosed infants. That is the same estimate reported in reference 22. According to Table 1 in reference 22, that analysis assumed that all infants with SMA have either 2 or 3 SMN2 copies. Are infants with 4 or more SMN2 copies classified as false positives in the NSW/ACT NBS program?

L192-195: The dominance of the combination of NBS and gene therapy relative to no NBS and nusinersen following onset of symptoms depends on a number of assumptions that should be made explicit. Most importantly, the authors assume a “one-and-done” gene therapy that eliminates the need for further treatment, whether with nusinersen or a repeated application of the gene therapy. That is an active topic of discussion. In the USA, many infants apparently have received both nusinersen and gene therapy. What has the Australian experience been?

L448: The list of references requires careful editing for consistency in formatting, e.g., sentence versus title capitalization. Some references are listed twice, with different reference numbers.

Author Response

Dear Reviewer,

Thank you for the comments that we have addressed in the attached DPF in blue texts. Please see the attachement.

Please note that line numbers may change in the revision from the previous version. We have made it clear as possible we can in the responses.

Kind Regards,

Sophy Shih

Reviewer 2 Report

The article, “Modelling the cost-effectiveness and budget impact of a newborn screening program for spinal muscular atrophy and severe combined immunodeficiency,” generates the first comparative health economic data based on a prior developed (i.e., companion model mentioned in manuscript) a newborn screening decision-analytic model that assess the economic value of severe combined immunodeficiency (SCID) NBS alone vs. a combined SCID and spinal muscular atrophy (SMA) NBS strategy. The study follows primary guidelines in conducting economic evaluation models and is likely of interest to a broad readership of IJNS.

The authors provide sufficient rationale for examining the cost-effectiveness of combined NBS of SMA and SCID, specifically that the incremental unit cost of adding SMA to the existing SCID panel is marginal. Accordingly, the reference case for combined SMA and SCID NBS is SCID only. However, stakeholders may be interested in the results of an economic evaluation that incorporates their representative environment where neither SMA or SCID is screened (INJS articles should appeal to international audiences). In addition, it may be the case that the SMA+SCID and neither-screened strategies are strictly dominant strategy-pairs and so SCID alone could be a suboptimal strategy. Alternatively, such a case can occur if we look at SMA screening alone. The authors note the inclusion of a companion paper in this issue, which presumably can be used to supplement the analysis and reporting in this manuscript. The results of this analysis should at minimum include the same base used in the companion analysis, and if possible, include an SMA alone screening strategy for identical reasons as above.

The authors report quality-adjusted life-years (QALYs) in their analysis. This reviewer is highly skeptical of the use of discounted QALYs as an effectiveness measure for newborn screening since we do not know if utility is well-defined among 0 to 17 year-olds—the parent-proxy approach that are derived for 5-17 year olds in the references used in this study also has foundational limitations. Back of the envelope calculations on how the study results would change if effectiveness measures such as discounted life-years saved or discounted adverse event-free life-years is used instead. If such calculations are not feasible, please provide additional discussion in limitations.

The study accounts for uncertainty in reporting the cost-effectiveness analysis as via the probability sensitivity analysis (PSA). The authors repost a very high probability of simulated results being cost-effective (97%). Raw data in Figures 4 and 5 illustrate strong positive correlation between the incremental cost and effectiveness pairs, which may or may not be consistent with real-world data since the correlation is determined by the model and not observed. Please note the limitation of the PSA results, and provide the following supplemental approach for calculating the PSA: using the monte carlo sample variance used for figures 4 and 5 and approximate (assuming ~normal) a 95% range separately for each incremental cost and effectiveness distributions and create ICER limits.

Budget impact analyses are less useful the further the time horizons. The assumption that all else is equal for a stakeholders budget is a strong limitation. It is valuable to provide the level of uncertaintuy for years 3,4, and 5 for stakeholders interested in expanding NBS. Please provide confidence intervals or a measure of variance for the budget impact results in table.

Minor comments

Screenshots of TreeAge decision schematics (line 170-171) and result diagrams (lines 290-291) that display the variables as they were titled in the authors software table input (e.g., c_NusinersonMaintenance (14,3175.05169 to 95,450.03446)”) is not ideal for publication. Please replace with more appropriate diagrams if published.

Line 42. Use of the term “Pan-ethnic” incidence rate is unusual.

Lines 93-94. Expand on the discussion on the findings of the studies that examined the cost-effectiveness of SMA screening. What were there principal findings?

Author Response

Dear Reviewer,

Thank you for the comments that we have addressed in the revision and response to the comments PDF in blue text. Please see the attachment.

Please note that line numbers may change in the revision from the previous version.

Kind Regards,

Sophy Shih

Round 2

Reviewer 1 Report

No further comments

Author Response

Dear Reviewer,

Thank you for your comments and report.

Kind Regards,

Sophy TF Shih